# Direct circRNA-mRNA Binding Controls mRNA Fate: A New Mechanism for circRNAs

**DOI:** 10.3390/ncrna11040053

**Published:** 2025-07-18

**Authors:** Raffaele Garraffo, Manuel Beltran Nebot

**Affiliations:** Department of Biology and Biotechnology Charles Darwin, Sapienza University of Rome, 00185 Rome, Italy

**Keywords:** circRNAs, mRNA regulation, RNA-RNA interaction

## Abstract

Circular RNAs (circRNAs) are covalently closed RNA molecules generated through a non-canonical splicing event known as back-splicing. This particular class of non-coding RNAs has attracted growing interest due to its evolutionary conservation across eukaryotes, high expression in the central nervous system, and frequent dysregulation in various pathological conditions, including cancer. Traditionally, circRNAs have been characterised by their ability to function as microRNA (miRNA) and protein sponges. However, recent discoveries from multiple research groups have uncovered a novel and potentially transformative mechanism of action: the direct interaction of circRNAs with messenger RNAs (mRNAs) to regulate their fate. These interactions can influence mRNA stability and translation, revealing a new layer of post-transcriptional gene regulation. In this review, we present and analyse the latest evidence supporting the emerging role of circRNAs in diverse biological contexts. We highlight the growing body of research demonstrating circRNA-mRNA interactions as a functional regulatory mechanism and explore their involvement in key physiological and pathophysiological processes. Understanding this novel mechanism expands our knowledge of RNA-based regulation and opens new opportunities for therapeutic strategies targeting circRNA-mRNA networks in human disease.

## 1. Introduction

Although most of the human genome is transcribed, less than 2% encodes proteins [1]. The vast majority of the transcriptome consists of non-coding RNAs (ncRNAs), which play diverse and crucial roles in cellular function. Over the past 30 years, the study of ncRNAs has grown into a major field of research. Thousands of ncRNAs have been identified and linked to various physiological and pathological processes. However, the molecular mechanisms underlying these functions often remain elusive or difficult to fully characterise [2,3,4]. A widely used approach to uncover the roles of ncRNAs involves identifying their molecular interaction partners and inferring their function through gain or loss-of-function experiments. This strategy has significantly advanced our understanding over the past two decades, revealing how many ncRNAs regulate key cellular processes [5].

Beyond protein interactions, as the classical studied partner of the RNAs, many ncRNA subtypes have been discovered to interact with other RNAs, by themselves or with the help of protein complexes. Classic examples of those interactions are mRNA with small nuclear RNA (snRNA), mRNA with microRNA (miRNA), small nucleolar RNA (snoRNA) with rRNA (ribosomal RNA) or tRNA (transfer RNA), long non-coding RNA (lncRNA) with microRNA (miRNA), and circular RNA (circRNA) with miRNA [6,7,8,9,10,11]. In recent years, significant advancements in both experimental techniques and computational pipelines have greatly improved our ability to characterise RNA-RNA interactions (RRIs) both inter- and intra-molecularly. RNA-binding proteins (RBPs) play a critical role in orchestrating the post-transcriptional regulation of gene expression by mediating a wide range of RNA-RNA interactions, which seems fundamental to numerous cellular processes, including splicing, translation, RNA transport, and stability [12]. To investigate these complex networks, several high-throughput techniques have been developed during the last decade. On one side are protein-centric methodologies such as CLASH, MARIO, RIC-seq, AGO-CLIP, hiCLIP, and PIP-seq, which have enabled researchers to systematically identify and characterise both intra- and inter-molecular RNA-RNA interactions [13,14,15,16,17,18]. These approaches typically rely on immunoprecipitation of ribonucleoproteins (RBPs) crosslinked to RNA, followed by ligation of physically proximate RNA fragments and sequencing, allowing the reconstruction of RNA duplexes associated with specific proteins. In parallel, RNA-centric strategies have emerged to provide a more direct and global view of RNA-RNA interactions across the transcriptome, independent of protein mediation. Techniques such as PARS, PARTE, FragSeq, SHAPE-seq, LIGR-seq, PARIS, and SPLASH leverage biochemical probing and crosslinking techniques to capture RNA duplexes in vivo or in vitro [19,20,21,22,23,24,25]. These methods often involve the application of crosslinking agents such as psoralen derivatives to stabilise native RNA-RNA interactions. This is followed by selective digestion of single-stranded RNA regions, proximity ligation of the remaining duplexed RNA ends, and high-throughput sequencing to identify base-paired regions and hybrid formations. Such approaches have significantly expanded our understanding of RNA structure and the dynamic nature of RNA interactomes.

Despite the remarkable progress facilitated by these methodologies and the identification of thousands of RNA-RNA interaction pairs across various cell types and organisms, our current map of the RNA-RNA interactome remains far from complete. Technical challenges—including correct hybrid detection, biases in detection of ribosomal RRIs due to their abundance, and the underrepresentation of inter-molecular RNA-RNA interactions—are still challenges to be resolved. Further development of more sensitive, unbiased, and high-resolution techniques and pipelines for detection will be essential to achieve a comprehensive and functional annotation of the RNA-RNA interactome and its role in cellular regulation and disease mechanisms [26].

Circular RNAs (circRNAs) are a diverse and abundant sub-family of non-coding RNAs characterised by their covalently closed loop structure. This unique circular conformation is generated through a non-canonical splicing process known as back-splicing, where a downstream splice donor site is joined to an upstream splice acceptor site [27,28,29,30,31]. Primarily discovered many years ago as a viral RNA or byproducts of splicing, circRNAs gained their due consideration with the popularisation of massive high-throughput sequencing and the wide usage of new specialised pipelines. Their importance became apparent because of their increased stability in confrontation with other ncRNAs, their high expression pattern in the central nervous system, and their deregulation in several physiopathological processes such as cancer [32,33,34]. These versatile molecules can regulate gene function in several ways depending on a variety of factors. For example, it has been described that nuclear circRNAs can interact with DNA to create R-loops and regulate genome stability [35,36]. However, most of the circRNAs are exported to the cytoplasm with a specific export machinery [37,38] where they can regulate gene expression through different mechanisms: some circRNAs bind microRNAs [39,40,41], possibly acting as sponges or as modulators of their stability and localisation, while others act as protein scaffolds [42] or as templates for translation [43,44]. However, those mechanisms of action might only be applied to specific cases due to stoichiometric issues in the case of the sponging mechanism or due to the low efficiency of IRES-mediated translation in the case of translatable circRNAs [45,46]. In recent years, a novel mechanism of action has been added to the functional repertoire of circular RNAs (circRNAs): gene expression regulation through circRNA-RNA interactions (Table 1).

This emerging mechanism, which involves direct or indirect binding between circRNAs and other RNA molecules, offers a potentially more efficient and less stoichiometry-dependent mode of regulation compared to traditional sponging circRNA-miRNA or circRNA-protein interactions. As this mechanism continues to be explored, it may become a cornerstone in circRNA research. In this text, we present a comprehensive overview of this rapidly expanding field and its implications for RNA biology.

## 2. *CircZNF609* Stabilises *CKAP5* mRNA Helping the Loading of ELAVL1 Protein

The earliest example of a circular RNA directly interacting with an mRNA can be found in the functional significance of the circular RNA *circZNF609* in cancer [47]. This study, primarily conducted in a rhabdomyosarcoma tumour model, was built upon previous findings from the same laboratory indicating the importance of this circRNA in rhabdomyosarcoma growth [48]. They employed psoralen crosslinking followed by specific RNA pull-down assays to identify mRNA interactors of *circZNF609*, successfully demonstrating and validating several in vivo circRNA-mRNA interactions. Interestingly, upon *circZNF609* knockdown, some of those target mRNAs exhibited reduced expression. The authors then dissected the molecular mechanism in one of these RNA-RNA pairs: *circZNF609-CKAP5* mRNA. They showed that this interaction enhances the ability of the RNA-binding protein ELAVL1 to bind *CKAP5* mRNA, thereby stabilising the transcript and increasing its translation [49,50]. They also provided a detailed characterisation of the RNA pairing, describing an approximately 80-nucleotide-long partially complementary region between the back-splicing junction of the circRNA and the 3′ UTR of the *CKAP5* mRNA. Notably, the researchers designed LNA-modified antisense oligonucleotides to disrupt this interaction and observed that this produced similar effects to *circZNF609* knockdown. Finally, the study demonstrated the physiological and pathological relevance of this novel regulatory mechanism. CKAP5, a microtubule polymerase, is essential for maintaining microtubule dynamics. The authors showed that disrupting the *circZNF609*-*CKAP5* mRNA interaction altered microtubule dynamics, impaired mitotic progression, reduced cell proliferation, and affected the cells’ sensitivity to chemotherapeutic agents targeting microtubules such as Vincristine. Importantly, they also demonstrated that this mechanism is conserved in other cancer cell types, including neuroblastoma and leukaemia cell lines, suggesting that circRNA-mRNA interactions may represent a general regulatory mechanism with potential relevance across multiple cancer contexts [47,51].

## 3. *CircHOMER1* Regulates Synaptic Localisation of *HOMER1b* mRNA Competing with ELAVL4 Protein

At the same time as the previous article, another study emerged highlighting the importance of circRNA-mRNA interactions, specifically within the neural system. This work [52] describes the mechanism of action of *circHomer1*, a circular RNA generated from the Homer1 locus, an essential gene involved in neural function and synaptic plasticity in humans [53]. The authors demonstrate a negative correlation between *circHomer1* and *Homer1b* mRNA in terms of their synaptic localisation within the orbitofrontal cortex (OFC) of control individuals compared to patients with bipolar disorder (BD) or schizophrenia (SCZ). Analysis of postmortem brain tissues from individuals with BD and SCZ revealed altered expression patterns of *circHomer1* and *Homer1b* in both the OFC and dorsolateral prefrontal cortex (DLPFC). Notably, circHomer1 was significantly downregulated in the DLPFC of SCZ patients and in the OFC of BD patients with a history of psychosis. These findings suggest that dysregulation of the *circHomer1-Homer1b* axis may contribute to the pathophysiology of these psychiatric conditions. Functionally, knocking down *circHomer1* results in increased synaptic localisation of Homer1b, while reducing Homer1b mRNA levels leads to elevated synaptic *circHomer1*. This indicates a bidirectional, competitive interaction that influences their respective synaptic distributions. Using predictive modelling and RNA antisense purification techniques, the researchers further discovered that *circHomer1* can directly bind to the 3′ untranslated region (UTR) of *Homer1b* mRNA. This approximately 40-nucleotide-long interaction occurs near sites recognised by the RNA-binding protein ELAVL4, which is known to regulate mRNA stability and localisation [54]. The authors propose that *circHomer1* competes with *Homer1b* for ELAVL4 binding, thereby modulating Homer1b’s synaptic localisation. Finally, in vivo experiments in mice showed that knocking down *circHomer1* in the OFC impairs reversal learning performance, indicating reduced cognitive flexibility. Conversely, *Homer1b* knockdown enhances reversal learning. Interestingly, simultaneous knockdown of both *circHomer1* and *Homer1b* restores reversal learning performance to baseline levels, underscoring their antagonistic roles in modulating this cognitive process. This study marks another cornerstone in the expanding field of neurophysiology, where circRNAs play crucial regulatory roles. It clearly demonstrates that circRNA-mRNA interactions may influence a wide range of physiological processes.

## 4. *CircARID1A* Enhances *SLC7A5* mRNA Stability by Scaffolding IGF2BP3

The next circRNA-mRNA interaction described involves *circArid1a* and *SLC7A5* mRNA. The authors characterise *circArid1a* as a circular RNA derived from exons 2, 3, and 4 of the Arid1a gene [55]. This cytoplasmic circRNA is found to be overexpressed in gastric cancer samples and appears to promote cell proliferation. Interestingly, they found out that RNA-FISH signals of *circARID1A* and *SLC7A5* matched with IF signal of IGF2BP3, whereas the colocalization was decreased upon *circARID1A* knockdown. Moreover, *SLC7A5* mRNA was enriched in *circARID1A* pull-down assay with respect to control probes, as well as in IGF2BP3 RNA immunoprecipitation (RIP) assay. Notably, the knockdown of *circARID1A* significantly reduced the enrichment of *SLC7A5* mRNA in IGF2BP3 RIP assay.

Altogether these results allowed the authors to propose a model in which circArid1a interacts with and facilitates the transfer of the IGF2BP3 protein to the *SLC7A5* mRNA, thereby enhancing its stability and translation.

The predicted interaction region spans approximately less than 100 nucleotides and involves both the circular RNA and the open reading frame (ORF) of the *SLC7A5* mRNA. Although this study provides less molecular evidence than previous reports, it nonetheless presents another compelling example of circRNA-mRNA interactions that may influence mRNA fate and contribute to pathophysiological processes.

## 5. CLiPPR-Seq Is a Genome-Wide Identification Technique That Detects Widespread circRNA–mRNA Interactions

This work marks a pivotal step forward in the understanding of circRNA-mRNA interactions by introducing a powerful, genome-wide approach to investigate these interactions systematically [56]. The authors developed CLiPPR-seq (Cross-Linking Poly(A) Pulldown RNase R Sequencing), a novel technique that combines psoralen-mediated crosslinking with RNA pulldown and sequencing to identify circular RNAs (circRNAs) bound to protein-coding mRNAs in living cells. This method relies on psoralen-induced covalent crosslinking of RNA duplexes, followed by mRNA enrichment. The samples are then treated with RNase R, an exonuclease that digests linear RNAs while preserving circular RNAs, thereby enabling the isolation of circRNAs that are physically associated with mRNAs. CLiPPR-seq was applied to three different cell lines: HeLa, βTC6, and C2C12. After rigorous filtering and analysis, the authors identified 1026 circRNA-mRNA interactions in βTC6, 518 in HeLa, and 448 in C2C12 cells. These results provide strong evidence that circRNA-mRNA interactions are not only widespread but may also have important regulatory implications. However, a limitation of the method is that it does not directly reveal specific circRNA-mRNA pairs; instead, it identifies them separately. To predict potential interactions, the authors used BLAST (BLAST+ suite) to search for complementary sequences between circRNAs and mRNAs. Several of these predicted interactions were experimentally validated using RNA pulldown assays and functional knockdowns. One compelling example is *circMtcl1a*, which was found to interact with multiple mRNAs, including *Trib2*, *Rps6Kc1*, and *2310039H08Rik*. Silencing *circMtcl1a* led to increased expression of *Rps6Kc1* and reduced expression of *2310039H08Rik*, supporting a functional role for circRNA-mRNA binding in gene regulation. Interestingly, the final effect of the interaction in the mRNA fate will depend on the other factors that might be regulating the mRNA, so in some cases, the circRNA-mRNA will provide stability and/or enhanced translation, and in other cases, the contrary. This work not only highlights the pervasiveness of circRNA-mRNA interactions but also establishes CLiPPR-seq as a foundational tool for exploring their roles in post-transcriptional regulation across the transcriptome.

## 6. *CircDLC1* Regulates *Gria1* and *Grin2a* mRNAs Translation Through Direct RNA–RNA Interaction and miRNA Competition

At the same time as the previous study, new pieces of evidence of circRNA-mRNA interactions in the central nervous system were appearing. A new study explored the complex regulatory role of circular RNAs (circRNAs) in neuronal signalling, specifically focusing on the involvement of *circDLC1* in glutamatergic neurotransmission [57]. Starting from the observation that mRNAs associated with glutamate receptor signalling (gluRNAs) were upregulated at a post-transcriptional level in a mouse model KO for *circDLC1*, the authors identified a novel tripartite interaction among the circRNA, gluRNAs, and a shared microRNA (miRNA) that collectively modulate gene expression relevant to excitatory signalling in the mouse brain. In fact, as they have demonstrated through circRNA pulldown, *circDLC1* interacts with *miR-130b-5p* as well as with 2 gluRNAs, *Gria1* and *Grin2a* mRNAs. Using imaging experiments and luciferase reporter assays, they described how *circDLC1* can regulate *miR-130b-5p* localisation acting synergically with the miRNA to repress Gria1 and Grin2a mRNAs translation and stability. In addition to its ability to regulate the availability of the miRNA, this circRNA can also regulate the translation of *Gria1* and *Grin2a* mRNAs via circRNA-mRNA interaction in a miRNA-independent way, as they have demonstrated by luciferase reporter assays. This dual function places the circRNA at the centre of a regulatory hub, finely tuning glutamatergic signalling pathways. Disruption of this interaction affects synaptic gene expression and alters neuronal activity, underlining its physiological relevance. The findings demonstrate that circRNAs can simultaneously regulate mRNA expression through miRNA localisation, expanding the current understanding of post-transcriptional regulation in the brain. The study also provides insight into how such regulatory networks might be leveraged for therapeutic strategies in neurological disorders involving excitatory/inhibitory imbalance. Overall, this work highlights a layer of gene expression control in neurons and establishes a framework for studying circRNA-centred regulatory networks that integrate multiple RNA species to control complex cellular functions such as synaptic signalling.

## 7. *CircHIPK3* Regulates *BRCA1* mRNA Translation Competing with FMRP

Another interesting example of a circular RNA (circRNA) with significant impact in a physiopathological context, specifically cancer, is the work on the functional characterisation of *circHIPK3* [58]. This circRNA, derived from the second exon of the HIPK3 gene, has been widely associated with cancer progression. While traditionally known for its role in miRNA sponging, this study highlights a novel mechanism involving direct circRNA-mRNA interaction, shifting the paradigm of circRNA functionality. Using RNA pull-down assays under psoralen crosslinking conditions to preserve native RNA-RNA interactions within the cell, and an extensive series of molecular biology experiments to demonstrate direct interactions between *circHIPK3*, *BRCA1* mRNA, and RNA-binding proteins, the authors described a new post-transcriptional regulatory axis involving *circHIPK3*, *BRCA1* mRNA, and the RNA-binding protein FMRP [59,60] (Fragile X Mental Retardation Protein). These results were supported by in vitro binding assays, including mutation analysis, and in vivo antisense oligonucleotide transfection and crosslinked RNA immunoprecipitation. Their results show that *circHIPK3* and FMRP competitively bind to *BRCA1* mRNA, modulating its stability and translation. Specifically, *circHIPK3* binding stabilises *BRCA1* mRNA and enhances its translation, thereby supporting a robust DNA damage response. In contrast, FMRP association leads to destabilisation and reduced translation of *BRCA1* mRNA. Again, this interaction was around 40 nucleotides between the back-splicing junction of the circRNA and a region close to the STOP codon in the ORF of the mRNA. This competitive interaction functions as a molecular switch, regulating BRCA1 protein levels and, consequently, the efficiency of DNA repair mechanisms. Functional assays further confirm that manipulating *circHIPK3* or FMRP levels significantly alters BRCA1 expression and the cell’s capacity to repair DNA damage, which has an important impact on genome stability and general cell homeostasis. Importantly, the authors employed modified antisense oligonucleotides to disrupt the *circHIPK3-BRCA1* mRNA interaction in vivo to regulate BRCA1 protein, altering DNA damage response and the general homeostasis of the cell. Again, the authors demonstrated that the circRNA-mRNA interaction is conserved in several tissues and cancer models, suggesting a general role of this mechanism in the body, more than a particular regulation in a specific cancer clone. These interventions not only validated the biological significance of this pathway but also demonstrated the therapeutic potential of targeting circRNA-mRNA interactions in cancer.

## 8. CircRNA-mRNA Pairings Induce Nonsense-Mediated Decay by Transferring Exon-Junction Complexes from circRNA to mRNA 3′ UTRs

In late 2024 a new keystone work appeared [61] with a whole new approach to decipher the function of circRNA-mRNA in gene regulation, in this case by promoting nonsense-mediated mRNA decay (NMD) [62,63,64]. Using an exogenous system, the researchers demonstrate that specific circRNAs can directly interact with complementary sequences in the 3′ untranslated regions (3′ UTRs) of target mRNAs, forming RNA-RNA duplexes. Molecularly, this interaction transfers exon-junction complexes (EJCs), placed onto circRNAs during back-splicing, in the 3′ UTRs of the mRNAs, thereby triggering EJC-dependent NMD, and degrading the mRNA in a process termed circRNA-induced NMD (circNMD). Using this model, the author finely details the number, size, and distance of the interaction to maximise the effect of this downregulation. Curiously, they describe this interaction as at least 21 nucleotides long and mostly complementary (only three mismatches). To research if these discoveries had an endogenous counterpart, the authors performed an immunoprecipitation of RNA attached to CBP80 and eIF4E (5′ CAP proteins), followed by a poly-A extraction and circular RNA identification in control cells and cells defective in NMD. Transcriptomic analyses successfully revealed hundreds of potential circNMD candidates and potential mRNAs regulated by NMD; however, the list did not include any information regarding which circRNAs were attached to which mRNAs. To obtain this information authors used data from in situ conformation sequencing (RIC-seq) experiments [18]. Combining this information together, the authors describe how circNMD of *BCL2L11* mRNA contributes to the regulation of cellular apoptosis [65,66]. However, these endogenous circRNA-mRNA interactions are less complementary than the exogenous models and look like the non-fully complementary pairings observed in the other works that were previously mentioned. Furthermore, the study shows that exogenously expressed circRNAs designed to interact with the 3′ UTRs of endogenous mRNAs can significantly downregulate mRNA levels, highlighting the potential therapeutic applications of circNMD in selective mRNA degradation. These findings provide compelling molecular evidence for circNMD, expanding our understanding of circRNA-mRNA functions and offering new avenues for gene regulation strategies.

## 9. *CircFOXK2* Promotes Stabilisation and Translation of *CCND1* mRNA Helping the Loading of ELAVL1 Protein

The last input in the field of the circRNA-mRNA interaction appeared few months ago, delineating a novel mechanism by which circRNAs promote tumour progression and endocrine resistance in oestrogen receptor-positive (ER+) breast cancer [67]. The study focuses on *circFOXK2*, a highly expressed circRNA in ER+ breast tumours, which was found to promote cell proliferation and confer resistance to endocrine therapies. Mechanistically, *circFOXK2* was predicted to pair directly with *cyclin D1* (*CCND1*) mRNA via complementary sequences in the circRNA and the 3′ UTR of the coding gene, giving an RNA duplex of around 40 nucleotides. To validate this pairing, an in vitro RNA-RNA interaction assay between these two sequences was performed, confirming the circRNA-mRNA interaction. On top of that, the depletion of the putative interacting nucleotides in *circFOXK2* significantly abrogates the interaction between the circRNA and the mRNA. As a further confirmation, a ChIRP-qPCR assay was performed to validate *circFOXK2*-*CCND1* mRNA interaction in vivo.

The authors demonstrated that this RNA-RNA duplex formation facilitates the loading of the RNA-binding protein ELAVL1 to the mRNA, which leads to enhanced stability and translation of *CCND1* mRNA. In fact, RNA immunoprecipitation (RIP) assay of ELAVL1 showed that the protein could interact both with the circRNA and the mRNA, and interestingly that the enrichment of *CCND1* mRNA was significantly reduced upon *circFOXK2* knockdown.

The regulation of CCND1 has an impact on the cell cycle and, of course, on the growth of the cell [68]. The functional relevance of this interaction was demonstrated through *circFOXK2* silencing and antisense oligonucleotide approaches, which led to reduced tumour growth and restored endocrine sensitivity in both in vitro models and xenograft mouse models.

**Table 1 ncrna-11-00053-t001:** Described circRNA-mRNA interactions.

Work	circRNA	Number of circRNA-mRNA Interactions	Methods
Rossi et al., 2022 [47]	*circZNF609*	11	Psoralen and native circRNA pull down, LNA ASO
Hafez et al., 2022 [52]	*circHOMER1*	2	Native circRNA pull down and RNA Antisense Purification
Ma et al., 2022 [55]	*circARID1A*	1	Native circRNA pull down
Singh et al., 2024 [56]	Several	+2000	CLIPPR-seq
Silenzi et al., 2024 [57]	*circDLC1*	2	Native circRNA pull down
Grelloni et al., 2024 [58]	*circHIPK3*	15	Psoralen and native circRNA pull down, LNA ASO
Boo et al., 2024 [61]	Several	41	RIC-seq, LNA ASO
Yi et al., 2025 [67]	*circFOXK2*	1	In vitro RNA–RNA interaction assay

## 10. Discussion

Over the past few decades, significant technical advancements have shifted the perception of non-coding RNAs (ncRNAs) from being considered mere transcriptional noise to being recognised as central regulatory hubs in cellular biology. As the number of identified ncRNAs has grown, so too has our understanding of their functional roles and the complex mechanisms they participate in. In recent years, technological leaps in methods involving psoralen crosslinking, proximity ligation, and high-throughput sequencing have enhanced our ability to study RNA-RNA interactions, leading to deeper insights into endogenous RNA networks. In the field of circular RNAs (circRNAs), the understanding of RNA-RNA interactions has progressed significantly. Initially, circRNAs were mainly studied for their interactions with microRNAs (miRNAs) in a sponging capacity, though the molecular details of these interactions were often poorly characterised. Now, a more detailed picture is emerging, highlighting complex interplays among circRNAs, mRNAs, and proteins that together regulate mRNA stability and fate.

This new mechanism entails a much more suitable stoichiometry than miRNA sponging. CircRNA are usually expressed in the range of dozens to hundreds of molecules per cell, but rarely exceeding this number, even in cancer cells where they are supposed to be overexpressed. MicroRNAs are expressed in much higher numbers per cell and usually they affect several mRNA targets, so to regulate effectively some mRNA their numbers might be altered substantially. To act as a sponge a circRNA might be expressed in higher numbers and it might be able to bind several microRNAs at the same time, conditions that rarely occur. On the other hand, most mRNAs are expressed in the range of hundreds to thousands of copies per cell [58], making a lot of them, in the same range of expression as circRNA and hence making this regulation stoichiometrically possible. Moreover, while this regulation does not “block” their interactions, a circRNA might interact with some mRNA, loading its protein cargo and moving to other mRNA, increasing the stochiometric range of this new mechanism of action.

Although this field is still in its early stages, several patterns have begun to emerge from pioneering studies (Table 2). Notably, the endogenous RNA-RNA interactions observed tend to be 30–80 nucleotides in length and exhibit partial, but not full, sequence complementarity. This level of complementarity appears sufficient to stabilise the interactions while avoiding the recruitment of RNA-binding proteins like Dicer or other components that would typically trigger an immunogenic response via recognition of double-stranded RNA (dsRNA). A key question for future research is whether this pattern of interaction is functionally sufficient and what specific features of circRNA-mRNA binding confer functionality without eliciting immune activation. It will also be important to investigate whether dsRNA-stabilising or dsRNA-modifying proteins, such as STAU1 or ADAR1/2, play a role in maintaining these interactions.

Another commonality observed is the putative region of interaction; most of the works predicted or demonstrated that the interaction in the mRNA was in the 3′ UTR region or around the ending codon of a sequence. One can hypothesise that this is because the 3′ UTR region of the transcripts is a signal hub where the different ribonucleoproteins and microRNAs interact to regulate the final translation of the protein, so if the circRNA has to regulate translation, it should be in this region where they must interact. Another possible explanation might be the possibility of less structured regions that allow better RNA-RNA interactions among two different RNAs.

Unfortunately, actual methods that are used to identify RNA-RNA interactions do not possess enough resolution to analyse transcription-wide circRNA-mRNA interactions. In fact, all the RRI detection techniques can detect mostly intra-molecular RNA-RNA interactions that compose RNA secondary structures, being overwhelming preponderant. Moreover, general RRI discovery methods based on proximity ligation hardly find any circRNA due to the difficulty of detecting back-splicing junctions in the chimeric reads detected by those experiments, mainly due to the short length of the chimeric arms and the difficulty of the software to detect back-splicing junctions in those chimeric reads. An increase in the arm length on the chimeric reads as well as a new or combined pipeline for BSJ detection might solve these problems in the future. Some authors use a combined approach using predictive pairing bioinformatic tools with in vitro validation, however, those approaches are yet to be tested for a high number of interactions. Considering how challenging is to identify the circRNAs that interact with mRNAs, establishing the correct circRNA-mRNA pairs and the precise interaction region can only be made for now with low-output molecular techniques, limiting the *bona fide* examples with a detailed molecular RNA interaction region. Therefore, it raises the need for the development of higher-resolution techniques that would allow us to specifically enrich the inter-molecular RNA-RNA interactions, as well as strengthening the bioinformatic tools to identify circRNAs-derived reads in RNA-seq data.

The functional outcomes of circRNA-mRNA interactions appear to be highly context-dependent (Figure 1, Table 2). In some cases, increased sequence complementarity or the loading of exon junction complexes onto the mRNA can trigger nonsense-mediated decay (NMD). In contrast, other interactions may sterically block destabilising proteins such as FMRP1 or facilitate the recruitment of stabilising proteins like ELAVL1, thereby enhancing mRNA stability and translation. Moreover, we should reconsider the classical sponging mechanism. Classically, circRNAs can modulate the availability of miRNAs and proteins through sponging effects, avoiding the accessibility of these factors to the mRNA. However, the introduction of circRNA-mRNA interactions adds another regulatory layer, potentially increasing the local availability of these factors near specific mRNAs.

Current data seems to demonstrate that circRNA-mRNA interaction is present in diverse cell lines and some of the mechanisms seem to be conserved across cell identities (such is the case of *circHIPK3* or *circZNF609*). However, parameters such as the relative expression levels and binding affinities of miRNAs, such as other RBPs that might regulate certain mRNA, should be taken into account when predicting the regulatory outcome of a specific interaction. Systematic analyses will be essential to determine whether these effects follow reproducible patterns depending on specific mRNA regulations.

The intricate crosstalk among circRNA-mRNA, circRNA-protein, and protein-mRNA interactions constitutes a finely tuned regulatory network that plays a crucial role in modulating protein expression. This multilayered interaction landscape allows for dynamic control of gene expression at both the transcriptional and translational levels. CircRNAs have gathered considerable attention in recent years as promising therapeutic agents. Their inherent stability, resulting from the absence of free 5′ and 3′ ends, renders them resistant to exonuclease-mediated degradation, making them attractive candidates for biomedical applications [69,70]. In addition to acting as molecular sponges for microRNAs and RNA-binding proteins, emerging evidence suggests that some circRNAs can be translated into functional peptides, further expanding their potential roles in cellular physiology and disease. Now, we can add the regulators of the mRNA to this myriad of functions.

Moreover, the relevance of circRNA-mRNA interactions is highlighted by experimental findings showing that these interactions can be selectively disrupted in cells using chemically modified antisense oligonucleotides (ASOs) [71]. Such targeted interventions have been shown to elicit specific phenotypic outcomes, thereby highlighting the regulatory significance of these interactions in gene expression pathways. The ability to perturb circRNA-mRNA pairing with high specificity opens up new possibilities for therapeutic modulation of disease-relevant genes. While ASO-based strategies offer a promising means of disrupting circRNA-mRNA interactions, their therapeutic application still faces key challenges, particularly related to off-target effects and targeted delivery. Notably, the tissue-specific expression patterns of many circRNAs present an opportunity: by designing ASOs against circRNAs that are uniquely or highly expressed specifically in certain tissues, it may be possible to enhance targeting specificity and reduce systemic side effects. However, sequence similarity with linear RNAs can still lead to unintended hybridisation. Therefore, further development of precise computational design tools and advanced delivery platforms will be essential to fully harness the therapeutic potential of these approaches.

One open question is whether circRNA-mRNA interactions are evolutionarily conserved. While some circRNAs and their interacting mRNAs are expressed across species, no studies have yet directly compared interaction pairs in orthologous systems. Comparative transcriptomics and structure-guided alignment strategies could help determine whether such regulatory interactions could represent conserved, lineage-specific, or contextually rewired features.

The mapping and characterisation of these circRNA-mRNA interactions not only deepen our understanding of post-transcriptional gene regulation but also provide a compelling rationale for using circRNAs as both therapeutic targets and tools. As research in this area continues to evolve, circRNAs are poised to play an interesting role in the development of novel RNA-based therapeutics [72].

## Figures and Tables

**Figure 1 ncrna-11-00053-f001:**
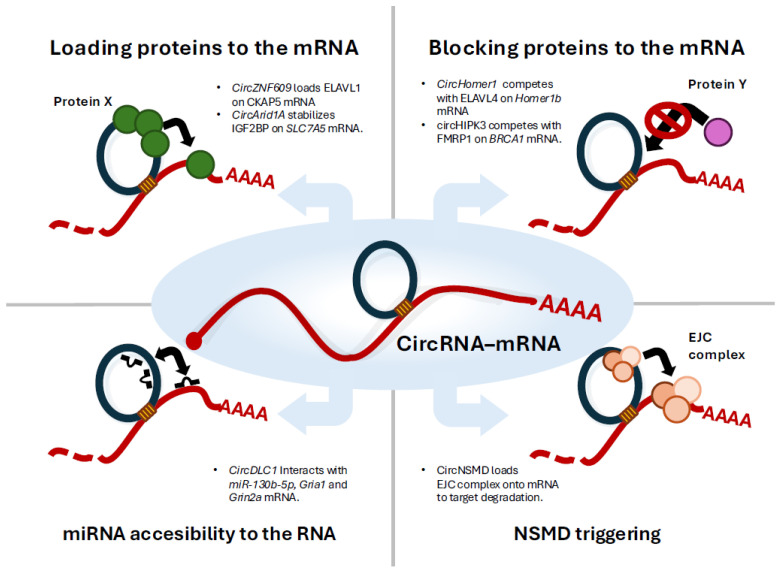
Cartoon showing the several mechanisms involving circRNA-mRNA interaction to regulate mRNA fate.

**Table 2 ncrna-11-00053-t002:** circRNA-mRNA characteristics.

Work	circRNA	mRNA Interactor	Effect on mRNA	Duplex Length (nt)	Pairing Region (mRNA)
Rossi et al. [47]	*circZNF609*	*CKAP5*	Positive	~60	3′ UTR
Hafez et al. [52]	*circHOMER1*	*HOMER1B*	Negative	~30	3′ UTR
Ma et al. [55]	*circARID1A*	*SLC7A5*	Positive	~90	CDS
Singh et al. [56]	*circACBD3*	*HPCA*	Negative	~20	3′ UTR
		*ARFGAP1*	Negative	~20	3′ UTR
	*circMTCL1*	*Rps6kc1*	Positive	~30	CDS
		*2310039H08Rik*	Negative	~20	CDS
Silenzi et al. [57]	circDLC1	*Gria1*	Negative	Not specified	3′ UTR
		*Grin2a*			
Grelloni et al. [58]	circHIPK3	*BRCA1*	Positive	~40	Last coding exon
Boo et al. [61]	circ_0002082	*BCL2L11*	Negative	~30	3′ UTR
	circ_0008496				
Yi et al. [67]	circFOXK2	*CCND1*	Positive	~80	3′ UTR

## Data Availability

This manuscript generated no new data. Any request of information can be requested from the corresponding author at manuel.beltrannebot@uniroma1.it.

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
