# Peer review of "Direct circRNA-mRNA Binding Controls mRNA Fate: A New Mechanism for circRNAs"

_ncrna, 2025, doi:10.3390/ncrna11040053_

Round 1
Reviewer 1 Report
Comments and Suggestions for Authors
Circular RNAs (circRNAs) formed by atypical splicing have emerged as essential regulators in eukaryotic cells in recent years. The current manuscript submitted by Garraffo and Nabot summarizes examples ot interactions between circRNA and mRNA that modify the expression of the mRNA in different ways. The manuscript is well-organized and provides a good overview through the inclusion of two tables and a schematic figure. The manuscript is thus of interest to a broad range of IJMS readers, particularly those working outside of circRNA research. However, I have several suggestions for improvements in the presentation.
- Add reference numbers to Tables 1 and 2.
- The inclusion of the history of contributing authors in the text is disturbing (author A did this, and then author 2 did that). I suggest focusing the text on experiments and conclusions, as this is the primary message the reader should take away. Include the author history in the tables by adding publication history to the tables.
- Line 78: “which lacks 5’ caps and 3’ poly(A) tails; this is self-evident since circular RNAs have no ends.
- Lines 116-119: complex phrase; try to simplify
- Please make sure all abbreviations are defined at first appearance.
Author Response
Circular RNAs (circRNAs) formed by atypical splicing have emerged as essential regulators in eukaryotic cells in recent years. The current manuscript submitted by Garraffo and Nabot summarizes examples ot interactions between circRNA and mRNA that modify the expression of the mRNA in different ways. The manuscript is well-organized and provides a good overview through the inclusion of two tables and a schematic figure. The manuscript is thus of interest to a broad range of IJMS readers, particularly those working outside of circRNA research. However, I have several suggestions for improvements in the presentation.
We thank reviewer for his comments to improve the quality of the manuscript, hereby a detailed answer to all his comments.
- Add reference numbers to Tables 1 and 2.
- We thank the reviewer for the suggestions. Reference numbers have been added to both Tables 1 and 2.
- The inclusion of the history of contributing authors in the text is disturbing (author A did this, and then author 2 did that). I suggest focusing the text on experiments and conclusions, as this is the primary message the reader should take away. Include the author history in the tables by adding publication history to the tables.
- We thank the reviewer for the thoughtful suggestion. We changed the main text as suggested by the author to stress the importance of the experiments. Line 110, 138, 171, 191, 221, 249, 382, 312.
- Line 78: “which lacks 5’ caps and 3’ poly(A) tails; this is self-evident since circular RNAs have no ends.
- We thank the reviewer for pointing it out. We have removed the sentence from the manuscript.
- Lines 116-119: complex phrase; try to simplify.
- The sentence has been revised and simplified to improve readability. Lines 115-117
- Please make sure all abbreviations are defined at first appearance.
- We thank the reviewer for the suggestion. We have carefully reviewed the manuscript to ensure that all abbreviations are defined upon first use. Unique exception represents the molecular techniques that has been described in the abbreviations sections to improve readability.
Reviewer 2 Report
Comments and Suggestions for Authors
Refer attachment

Author Response
The present review focuses on circRNAs, which directly bind to mRNAs to regulate their stability and translation, offering a novel layer of post-transcriptional gene regulation with implications for both physiological processes and diseases like cancer. This emerging mechanism, supported by recent studies, highlights circRNAs as key players in RNA-based regulatory networks and potential therapeutic targets.
We thank the reviewer for his comments and share his point of view on the key questions that must address the field. This work is intended to be a bibliographic review, so we were constrained by the impossibility to add new data other than the one is already published and peer reviewed. However, we adapted the manuscript to include his suggestions as far as possible.
Major Comments
- Some circRNA-mRNA interactions lack detailed molecular evidence (e.g., structural insights, binding kinetics, etc.). Also, please strengthen experimental validation (e.g., mutational analyses, in vitro binding assays), this would enhance mechanistic understanding.
We kindly thank the reviewer for the suggestion. We have added more experimental detail to several sections to clarify the level of mechanistic evidence supporting each study. Unfortunately, the quality of the molecular details differs in each article and were not available in the original works reviewed, we have acknowledged these limitations where relevant. Lines 174-179, 223-234, 259-260, 319-330.
- Current methods (e.g., CLIPPR-seq, RIC-seq) struggle with identifying exact circRNA-mRNA pairs and interaction sites. Authors should provide a discussion on developing higher-resolution techniques or combining complementary approaches to improve accuracy.
We appreciate the reviewer’s suggestion. A new paragraph has been added in the Discussion highlighting the limitations of current methods and proposing strategies to improve resolution and accuracy. Lines 384-402
- The functional outcomes of circRNA-mRNA interactions (e.g., mRNA stabilization vs decay) appear context-dependent. More systematic studies across cell types and conditions are to be included to define generalizable rules.
We could not agree more with the reviewer #2, and we already stated this important point in the discussion; more systematic studies are necessary to describe better the outcome of these interactions. We extended the paragraph in the discussion detailing the importance of the context for the functional outcome. Lines 417-423.
- While ASOs show promise in disrupting circRNA-mRNA pairs, off-target effects and delivery challenges remain. Therefore, please address these hurdles to support translational potential.
We thank the reviewer for pointing this out. We have added a section in the discussion to further analyse the possible issues arose from ASOs treatments. Lines 443-452.
- The conservation of circRNA-mRNA interactions across species is not much explored. Comparative analyses could reveal whether these mechanisms are broadly relevant or lineage-specific. Please include a detailed section outlining this.
We acknowledge the lack of data regarding this matter. We have added a note in the discussion to raise this open question. Lines 453-458.
- The review states stoichiometric limits in miRNA sponging but does not address how circRNA-mRNA binding efficiency compares quantitatively. Please add brief discussion.
We thank the reviewer for this insightful point. We added a paragraph in the discussion Lines 350-362.
Minor Comments
- Some terms (e.g., ‘sponging’ vs ‘sequestering’) are used interchangeably. Please standardize terminology.
We have revised the manuscript to ensure consistent use of terminology throughout.
- Figure 1 could benefit from labels or legends to explicitly link mechanisms to examples in the text (e.g., circHIPK3-FMRP competition). Figure is blurry and text is not readable.
We apologise for the lack of clarity of the figure; we uploaded a high-resolution image of the figure to the server but there are limitations in the figure conversion in the MDPI server. We reuploaded the figure again high-resolution, please let us know if the problems persist or contact the editor for high resolution images. We extended the labels in the figure.
- Inclusion of more figures would be beneficial.
We appreciate the suggestion. However, we wanted to synthetise the knowledge in one single figure to have a clear image of all the information present now in the field. We agree that is impossible to put all the information in one figure, and extra figures would help to make clearer the mechanism, but we decided to be concise and synthetic rather than descriptive.
Remark
Addressing these concerns will strengthen the review’s impact on both basic research and clinical applications.
Reviewer 3 Report
Comments and Suggestions for Authors
This is a well-structured and timely review on the emerging role of circRNA-mRNA interactions in post-transcriptional gene regulation. The manuscript comprehensively presents both mechanistic insights and experimental methodologies across various biological systems. The strength of the manuscript rely on the exhaustive and up-to-date literature survey, the balanced focus on both methods and findings and the clear tables summarizing interactions and mechanisms. Here are my minor comments to improve the quality of the manuscript:
- The manuscript is generally well-written, but many sections contain long, complex sentences that may hinder comprehension. For instance, lines like: "Using the combination of mouse models and molecular biology techniques, they uncover that this circRNA not only regulates the availability of the miRNA through a non-canonical mechanism..."
could be simplified to enhance readability. Some sections (e.g. the Introduction and Discussion) would benefit from more concise summarization of existing methods or findings. - The Abstract is strong, but could briefly mention specific examples (e.g., circZNF609 or circFOXK2) to make it more concrete. Section titles like “CircZNF609” or “CircHIPK3” might be better framed more descriptively, e.g., “CircZNF609 regulates microtubule dynamics via mRNA stabilization”,
- There is an inconsistent level of detail between studies. Some (like circZNF609 and circHIPK3) are extensively described with molecular detail, while others (like circARID1A) are quite brief. Try to balance the level of detail across studies.
- Table 1 and 2 are informative. Consider highlighting common themes visually, such as: Type of effect (positive/negative), Interaction length range, mRNA region (5', cds, 3'UTR)
- The manuscript does an excellent job of synthesizing recent discoveries. However, the novelty and implications could be further emphasized in the final Discussion, particularly: How circRNA-mRNA interactions differ from classical miRNA regulation. Why these mechanisms are more therapeutically viable (e.g., less stoichiometry-dependent)
The manuscript is generally well-written, but many sections contain long, complex sentences that may hinder comprehension. For instance, lines like: "Using the combination of mouse models and molecular biology techniques, they uncover that this circRNA not only regulates the availability of the miRNA through a non-canonical mechanism..."
could be simplified to enhance readability. Some sections (e.g. the Introduction and Discussion) would benefit from more concise summarization of existing methods or findings.
Author Response
This is a well-structured and timely review on the emerging role of circRNA-mRNA interactions in post-transcriptional gene regulation. The manuscript comprehensively presents both mechanistic insights and experimental methodologies across various biological systems. The strength of the manuscript rely on the exhaustive and up-to-date literature survey, the balanced focus on both methods and findings and the clear tables summarizing interactions and mechanisms.
We kindly thank the reviewer for his positive comments and suggestions to improve the manuscript, here a detailed list of the answers to his remarks.
Here are my minor comments to improve the quality of the manuscript:
- The manuscript is generally well-written, but many sections contain long, complex sentences that may hinder comprehension. For instance, lines like: "Using the combination of mouse models and molecular biology techniques, they uncover that this circRNA not only regulates the availability of the miRNA through a non-canonical mechanism..."
could be simplified to enhance readability. Some sections (e.g. the Introduction and Discussion) would benefit from more concise summarization of existing methods or findings. - We kindly thank the reviewer for the comment. We have proceeded to simplify several complex sentences, hoping that now the review results more clear and readable.
We tried to make a more concise discussion; however other reviewers pointed towards more complex explanations of the standpoints and implications of the circRNA-mRNA interactions. We hope this new version can satisfy both expectations and can summarize this complex field in a satisfactory manner. Citing the reviwer example 223-235.
- The Abstract is strong, but could briefly mention specific examples (e.g., circZNF609 or circFOXK2) to make it more concrete. Section titles like “CircZNF609” or “CircHIPK3” might be better framed more descriptively, e.g., “CircZNF609 regulates microtubule dynamics via mRNA stabilization”,
- We kindly thank the reviewer for the suggestions. We have changed the titles of the paragraph to be more informative; nevertheless, while we recognize the benefit of concrete examples, we chose to keep the Abstract general to maintain a broad perspective on circRNA-mRNA mechanisms.
- There is an inconsistent level of detail between studies. Some (like circZNF609 and circHIPK3) are extensively described with molecular detail, while others (like circARID1A) are quite brief. Try to balance the level of detail across studies.
- We thank the reviewer for the consideration. We added more information in those paragraphs less described, aiming for more uniform depth across all sections, however some of the articles only provide few experiments to demonstrate circRNA-mRNA interaction. We have stated this uneven distribution in the article.
- Table 1 and 2 are informative. Consider highlighting common themes visually, such as: Type of effect (positive/negative), Interaction length range, mRNA region (5', cds, 3'UTR)
- We thank to the reviewer for the suggestion. Unfortunately, we are limited by the editorial formatting policies and constrains of the journal where colours in the tables are not encouraged. We hope this point might not represent a major issue.
- The manuscript does an excellent job of synthesizing recent discoveries. However, the novelty and implications could be further emphasized in the final Discussion, particularly: How circRNA-mRNA interactions differ from classical miRNA regulation. Why these mechanisms are more therapeutically viable (e.g., less stoichiometry-dependent).
- We thank the reviewer for the suggestion. We have added some lines in the discussion to stress out the putative therapeutic advantages of targeting circRNA-mRNA interactions respect to other RNA-RNA interactions. Lines 443-458, 350-362.
The manuscript is generally well-written, but many sections contain long, complex sentences that may hinder comprehension. For instance, lines like: "Using the combination of mouse models and molecular biology techniques, they uncover that this circRNA not only regulates the availability of the miRNA through a non-canonical mechanism..."
could be simplified to enhance readability. Some sections (e.g. the Introduction and Discussion) would benefit from more concise summarization of existing methods or findings.
We are grateful for the reviewer’s insightful feedback, which has significantly improved the clarity and impact of our manuscript. We tried to make a more concise discussion; however other reviewers pointed towards more complex explanations of the standpoints and implications of the circRNA-mRNA interactions. We hope this new version can satisfy both expectations and can summarize this complex field in a satisfactory manner.